# Assessment of Microvascular Hemodynamic Adaptations in Finger Flexors of Climbers

**DOI:** 10.3390/bioengineering11040401

**Published:** 2024-04-19

**Authors:** Blai Ferrer-Uris, Albert Busquets, Faruk Beslija, Turgut Durduran

**Affiliations:** 1Institut Nacional d’Educació Física de Catalunya (INEFC), Universitat de Barcelona (UB), 08038 Barcelona, Spain; albert.busquets@gencat.cat; 2ICFO-Institut de Ciències Fotòniques, Barcelona Institute of Science and Technology (BIST), 08860 Castelldefels, Spain; faruk.beslija@icfo.eu (F.B.); turgut.durduran@icfo.eu (T.D.); 3Institució Catalana de Recerca i Estudis Avançats (ICREA), 08010 Barcelona, Spain

**Keywords:** rock climbing, sport climbing, near-infrared spectroscopy, diffuse correlation spectroscopy, endothelial function, finger flexors, vascular occlusion test

## Abstract

Climbing performance is greatly dependent on the endurance of the finger flexors which, in turn, depends on the ability to deliver and use oxygen within the muscle. Near-infrared spectroscopy (NIRS) and diffuse correlation spectroscopy (DCS) have provided new possibilities to explore these phenomena in the microvascular environment. The aim of the present study was to explore climbing-related microvascular adaptations through the comparison of the oxygen concentration and hemodynamics of the forearm between climbers and non-climber active individuals during a vascular occlusion test (VOT). Seventeen climbers and fifteen non-climbers joined the study. Through NIRS and DCS, the oxyhemoglobin (O_2_Hb) and deoxyhemoglobin (HHb) concentrations, tissue saturation index (TSI), and blood flow index (BFI) were obtained from the flexor digitorum profundus during the VOT. During the reactive hyperemia, climbers presented greater blood flow slopes (*p* = 0.043, *d* = 0.573), as well as greater O_2_Hb maximum values (*p* = 0.001, *d* = 1.263) and HHb minimum values (*p* = 0.009, *d* = 0.998), than non-climbers. The superior hemodynamics presented by climbers could indicate potential training-induced structural and functional adaptations that could enhance oxygen transportation to the muscle, and thus enhance muscle endurance and climbing performance.

## 1. Introduction

Repetitive intermittent forceful contractions of the finger flexor muscles are typically required to hold and progress through climbing routes. The ability to resist fatigue during this kind of contraction is thought to be one of the main performance factors in rock climbing [1,2]. This is especially true for the endurance of the flexor digitorum profundus (FDP), which is responsible for performing the flexion of the distal phalanxes of the fingers and is highly engaged in the main climbing grip positions [3]. When muscle activity is required, energetic demands occur and the muscle tissue needs to consume oxygen to meet these demands [4]. Moreover, intramuscular pressure rises during the muscle action and impairs blood flow [5,6]. Impairment of the blood flow causes a lack of oxygen transportation to the muscle, which along with the increased oxygen consumption causes a reduced O_2_ pressure (pO_2_) in the tissue [7]. Reduced pO_2_ stimulates the release of vasodilators, such as nitric oxide, which contribute to micovasculature vasodilation in order to increase blood flow. This, in turn, enhances O_2_ supply to the tissue, aiding in maintaining a sufficient energy supply through oxidative metabolism. Therefore, the ability to transport and consume oxygen within the muscle tissue might play a key role in the maintenance of a constant and sufficient energy supply to the muscle and preventing fatigue. In fact, previous research has pointed at these two factors as some of the key performance traits of advanced and elite climbers [2,8].

Previous research has shown that the ability to transport oxygen at the conduit artery level (macrovasculature) is similar between the trained and untrained healthy population [9]. In addition, macrovascular function is not associated with rock climbing performance [8]. Therefore, if no relevant distinctive adaptations in the macrovasculature can be observed in trained individuals, the key adaptations to muscle endurance may be found lower in the vascular downstream and within the muscle tissue itself. In fact, it is well-known that repeated muscle contractions (i.e., training) play an important role in the biogenesis of mitochondria, which are responsible for the oxidative metabolism and the synthesis of adenosine triphosphate (ATP) within the muscle [10]. At the same time, during repeated muscle contractions, the aforementioned increase in intramuscular pressure requires the activation of vasodilatoy mechanisms to increase blood flow and meet the imposed O_2_ requirements. As a consequence of the increased blood flow, greater transmural pressure and shear stress on the vascular walls will occur [7,11]. Repeated episodes of shear stress are thought to be the primary physiological signal for the structural (e.g., angiogenesis and vessel diameter) and functional (e.g., vasodilatory response capabilities) adaptation of the microvasculature [7,11]. Therefore, the usual intermittent isometric contractions of the finger flexors performed by climbers could trigger structural and functional adaptations of microcirculatory vessels and mitochondrial biogenesis, thereby enhancing finger flexor resistance to fatigue by improving their O_2_ supply and consumption capabilities, respectively. Vascular occlusion tests (VOT) have been used as an efficient and comfortable method to explore microvasculature function under resting conditions [12,13,14,15]. During VOT, a local ischemic environment is produced in the muscle tissue by placing and inflating a pressure cuff or a tourniquet at the proximal end of the limb and restricting blood flow for a certain period of time. During ischemia, the extent and rate of oxygen consumption within the tissue can provide valuable information regarding the oxidative metabolism. When the tourniquet is released, a reactive hyperemia is usually experienced, caused by the vasodilatory response of the microvasculature to the diminished pO_2_, the metabolite accumulation, and the shear stress that the flowing blood causes on the endothelium [16,17]. This endothelial function is thought to play the biggest role in such vasodilatory capacity, which in turn is related to the capacity to transport oxygen [9,18].

Near-infrared spectroscopy (NIRS) is a well-established technique to study the microvasculature non-invasively. NIRS explores muscle tissue oxygenation based on the absorption and scattering of near-infrared light when penetrating into the biological tissue [19]. Via NIRS, we can estimate changes in the microvascular concentration of oxyhemoglobin (O_2_Hb) and deoxyhemoglobin (HHb) in the skeletal muscle. Based on both concentration changes, the local oxygenation of the muscle can be characterized by the tissue saturation index (TSI), calculated as the percentage ratio between the O_2_Hb and the total hemoglobin (O_2_Hb + HHb) [19]. Note that TSI is specific to a certain series of NIRS oximeters by a particular brand, and other variables related to the microvascular blood oxygen saturation can be found in the literature.

Previous research using NIRS typically explored muscle oxygen consumption during a VOT by assessing the downslope dynamics of the TSI during the ischemia part of the test and the vasodilatory capabilities by characterizing the upslope dynamics of the TSI during the reactive hyperemia [15]. If we focus on the climbing population, previous studies observed specific adaptations in the forearm muscles during a VOT mainly during the reactive hyperemia. Fryer et al. [20] found that the time needed to reach halfway to the baseline level of the TSI during the reactive hyperemia (i.e., half-time to recovery, HTR) was greater for non-climber active participants than for climbers, suggesting that climbers possessed a greater capacity to recover from ischemia. Furthermore, it has also been observed that HTR is inversely associated with climbing performance, proposing a regressive model to predict climbing performance based on the HTR [21,22].

Despite the extended usage and relevance of the TSI and other indices of the tissue oxygen saturation status (like StO_2_ or SmO_2_) in the clinical setting, it is important to note that their calculation is highly affected by changes in blood volume. Saturation indices are easily interpreted during blood flow occlusion, when the blood volume is constant and thus changes in saturation are mainly related to oxygen consumption by the tissue. However, the interpretation of saturation indices might represent an issue during fluctuating blood volume conditions, like reactive hyperemia. During hyperemia, arteriolar vasodilation induced by ischemia or thermoregulation [4,23] causes a rapid and great increase in blood flow and blood volume, which goes far beyond the oxygen consumption of the muscle and makes the interpretation of saturation indices difficult. Therefore, contemplating O_2_Hb and HHb concentrations individually could help with hyperemia interpretation. Moreover, as suggested by previous research, the measurement of microvascular blood flow could provide a more direct, sensitive and meaningful view of changes in blood volume during hyperemia [4].

Although technical limitations have made it difficult to implement these kinds of measures in previous research, a novel non-invasive photonic technique called diffuse correlation spectroscopy (DCS) has been successfully applied to provide an index of microvascular blood flow (BFI) of the brain and skeletal muscle microvasculature [24]. DCS utilizes speckle fluctuations in near-infrared light patterns caused by moving scatterers (i.e., red blood cells) to assess blood flow with high temporal resolution [24]. Therefore, DCS constitutes a relevant tool to measure microvascular blood flow adaptations at rest. DCS has not yet been utilized extensively on muscles but it has been validated against gold standards, and its limitations (mainly due to motion sensitivity) have been described [25,26]. Furthermore, DCS has received increasing interest over the past few decades and has been successfully utilized in various clinical scenarios, like during rest or exercise, and with various populations, from healthy individuals to patients with peripheral arterial disease [27,28]. However, to our knowledge, DCS has not been used to assess differences between trained and untrained individuals and certainly it has not been reported in climbing research.

Therefore, the aim of the present study was to explore training-related microvascular adaptations through the comparison of oxygen concentration (O_2_Hb, HHb, and TSI) and hemodynamics (BFI) of the forearm between climbers and non-climber active individuals during a VOT.

## 2. Materials and Methods

### 2.1. Participants

Thirty-two participants (15 healthy physically active non-climbers and 17 healthy climbers) with no upper limb injuries during the last six months participated in the study (Table 1). The inclusion criteria for the climbing group were as follows: (a) minimum climbing experience of one year, (b) minimum of one climbing/training session per week, and (c) minimum self-rated ability (best climbing grade achieved) of 7a+ (French grading system) during the last six months. Climbing grades were converted to the International Rock Climbing Research Association’s (IRCRA) standardized metric scale for analysis purposes [29]. Participants’ written consent was obtained before the study initiation. The study was approved by the Ethics Committee of the local administration (approval code: 002/CEICGC/2021).

### 2.2. Procedures

#### 2.2.1. Experimental Protocol

All participants were asked to avoid climbing or high-intensity exercise for 24 h before the experimental session. Likewise, participants were asked to avoid caffeine and food consumption for two hours before the experimental session. Upon reporting to the laboratory, participants filled in the informed consent form and a self-reported health and sport-experience form. Afterwards, anthropometric measurements of height, weight, maximum forearm perimeter, and forearm skinfold were taken. Forearm skinfold was measured at the place of the probe location to ensure adequate penetration of the infrared light (see Table 1). Next, participants were asked to lie down on a massage therapy bed and rest in a supine position for 17 min before commencing the VOT. During this time, we dedicated 15 min to placing the NIRS and DCS probes over the belly of the flexor digitorum profundus (FDP) of the participant’s dominant arm and also to fitting them with a brachial artery tourniquet, located as close as possible to the brachial plexus. After this, a two-minute warm-up period for the NIRS was guaranteed before commencing the VOT.

#### 2.2.2. Data Collection

A continuous-wave NIRS device (Portalite, Artinis Medical Systems BV, Zetten, The Netherlands) was used. This NIRS device incorporated three light-emitting diodes positioned at 30, 35, and 40 mm from a single receiver. The diodes transmitted light in two wavelengths (760 and 850 nm) and data were acquired with a frequency of 50 Hz. It is important to note that NIRS cannot distinguish between hemoglobin and myoglobin; for brevity, we will refer to both as hemoglobin (Hb). The NIRS probe was located on the FDP muscle, over the line between the medial epicondyle of the hummer and the carpal lunate, proximally at 1/3 of the distance between the two landmarks [22]. The probe was secured with double-side adhesive tape and covered with an opaque black cloth loosely wrapped around the participant’s forearm in order to prevent ambient light from altering signal quality. The NIRS device provided measures of arbitrary concentration of oxyhemoglobin (O_2_Hb) and deoxyhemoglobin (HHb). From O_2_Hb and HHb, the tissue saturation index (TSI) was computed as the percentage ratio between O_2_Hb and O_2_Hb + HHb [19].

A custom-made DCS system (ICFO, Barcelona, Spain) was used to measure the blood flow with a sample frequency around 38.5 Hz. The system comprised a single source-detector measurement channel at 785 nm wavelength, with the source–detector separation of either 2.2 or 2.55 cm, depending on the forearm skinfold measurement, to ensure the sensitivity of the measurement to the muscle tissue. The probe was located on the FDP immediately above (distal) the NIRS probe location, with sources placed on the opposite sides, to eliminate possible crosstalk between the two measurements. It is important to acknowledge that this is a continuous measurement. Therefore, to fit for the absolute value of the blood flow index (BFI), the information on the absorption and the scattering should be known a priori. Since we did not have these values readily available for each subject, we were only able to retrieve the information on the relative changes in BFI of the participants.

#### 2.2.3. Vascular Occlusion Test

The VOT was composed of three distinct phases: baseline (BA), occlusion (OC) and reactive hyperemia (HY). BA measurements were performed for three minutes before inflating the pressure cuff (pressure 0 mmHg). At the end of these three minutes, participants were instructed to perform a light (~10% of their estimated maximum) contraction by squeezing a cylindrical object with their instrumented hand for five seconds. This light contraction sought to lightly stimulate the metabolic oxidation of the finger flexors, as has been performed in other studies [20,21,22,30]. The OC phase started with a rapid inflation of the cuff to a pressure considered over the limb occlusion pressure (220 mmHg), and then cuff pressure was maintained until TSI values stabilized for a period of 30 s, as has been performed in most of the preceding similar climbing studies [20,21,22,30]. This stabilization normally occurred after between 3 and 5 min, as has been reported in preceding publications [20,21,22,30]. When TSI stabilized, the cuff was rapidly deflated and the reactive hyperemia and post-occlusion changes were recorded for three minutes. During VOT, oxygen concentration changes and blood flow index values were continuously registered via NIRS and DCS, respectively.

### 2.3. Data Processing and Variables

A low pass filter with 1 Hz cut-off frequency, 60 dB stopband, and 0.85 steepness was applied to the NIRS and DCS data to remove the components related to the heart rate of the subjects. All DCS data were then averaged to 1 Hz to minimize the effects of the noise and the large fluctuations of BFI due to the cardiac cycle.

From these processed data, mean baseline values of the O_2_Hb, HHb, TSI, and BFI (BA-O_2_Hb, BA-HHb, BA-TSI, and BA-BFI) were obtained from 1 min of the BA phase of the VOT (Figure 1). These baseline values were utilized to compute delta values for the NIRS signals (∆O_2_Hb, ∆HHb, and ∆TSI) in the OC and HY phases. Similarly, BA-BFI was used to compute relative blood flow (rBF) values, thus expressing BFI as percentage of BA-BFI (rBF = BFI/BA-BFI × 100) in the OC and HY phases. For the NIRS signals during the OC, the linear slope of the initial 20 s of the occlusion after the tourniquet inflation (OC-∆O_2_Hb_slope_, OC-∆HHb_slope_, and OC-∆TSI_slope_), the occlusion peak value (OC-∆O_2_Hb_min_, OC-∆HHb_max_, and OC-∆TSI_min_), and the time-to-peak value (OC-∆O_2_Hb_tmin_, OC-∆HHb_tmax_, and OC-∆TSI_tmin_) were calculated (Figure 1). The minimum value of rBF during OC (OC-rBF_min_) was computed as the mean of the last 10 s before the release of the tourniquet (Figure 1). During the HY phase, upon tourniquet deflation, we computed the peak post-occlusive hyperaemic response value (HY-∆O_2_Hb_max_, HY-∆HHb_min_, HY-∆TSI_max_, and HY-rBF_max_), the time-to-peak hyperaemia (HY-∆O_2_Hb_tmax_, HY-∆HHb_tmin_, HY-∆TSI_tmax_, HY-rBF_tmax_), the linear slope of the 10 initial seconds after tourniquet release for the NIRS signals (HY-∆O_2_Hb_slope_, HY-∆HHb_slope_, and HY-∆TSI _slope_), the linear slope from tourniquet release to peak values for the rBF (HY-rBF_slope_), and the elapsed time from the tourniquet release to half of the height of the hyperemic response (HY-∆O_2_Hb_HTR_, HY-∆HHb_HTR_, HY-∆TSI_HTR_, HY-rBF_HTR_) (Figure 1).

### 2.4. Statistical Analysis

Data normality was checked via the exploration of histograms, Q–Q plots and Shapiro–Wilk’s test. Variable transformation was used when necessary. In order to control for the number of planned pairwise comparisons, main group effects during the OC and HY phases were explored through three multivariate Hotelling’s T^2^ tests: NIRS variables during the OC phase, NIRS variables during the HY phase, and DCS variables during the HY phase. Pairwise comparisons with Bonferroni’s correction were also conducted for individual variable analysis. Since we had a single variable from DCS signal during the OC phase, OC-rBF_min_, group differences were explored through Student’s *t*-test. In the case of non-normal variables, alternative non-parametric tests were utilized. After Bonferroni’s correction, the level of significance was set at *p* ≤ 0.05. Hotelling’s T^2^ test effect sizes were measured via partial eta squared (small effect size: *η*^2^*p* ≤ 0.010; medium effect size: *η*^2^*p* ≤ 0.059; large effect size: *η*^2^*p* ≤ 0.138). Cohen’s *d* effect sizes were calculated for pairwise comparisons according to Cohen [31] and interpreted as 0.2 representing a small, 0.5 representing a medium, and 0.8 representing a large effect.

## 3. Results

Participants from both groups showed a typical response to the execution of the VOT (see Figure 2 with example signals from one participant of each group). This means that they presented a stable BA phase, followed by an expected drop in blood flow and nearly constant oxygen consumption during the OC phase, and lastly a fast re-establishment of the blood flow and oxygen concentration raise at the start of the HY phase. Group means and standard deviations along with statistical results from the pairwise comparisons are presented in Table 2.

Logarithmic variable transformation was used for OC-rBF_min_, HY-rBF_max_, HY-∆TSI_max_, and HY-rBF_slope_ to fit the normal distribution of the data. The HY-rBF_HTR_ parameter could not be transformed and the Mann–Whitney U test was utilized for this variable. When Hotelling’s T^2^ tests were performed, we did not observe any main group effect during the OC phase of the test for NIRS variables (*F* (1, 30) = 1.445, *p* = 0.230, *η*^2^*p* = 0.371, power = 0.523). These results indicate that the rate and extent of oxygen consumption and the magnitude of the blood flow occlusion during the OC phase of the VOT were not influenced by climbing training. Along with the absence of main group effects, we did not observe any differences in the pairwise comparisons performed during the OC phase for any of the computed variables (Table 2).

Contrarily, a main group effect was found for the NIRS-related variables during the HY phase (*F* (1, 30) = 2.692, *p* = 0.027, *η*^2^*p* = 0.660, power = 0.868), indicating that groups were different when compared through the compound of variables obtained from the NIRS signals. When pairwise comparisons were explored, we found that climbers presented greater HY-∆O_2_Hb_max_ values (*p* = 0.001, *d* = 1.263) and lower HY-∆HHb_min_ values (*p* = 0.009, *d* = 0.998) than non-climbers (Figure 3 and as can been seen in Figure 2 with sample data from each group). Despite differences in the magnitude of O_2_Hb and HHb reactions, no differences were found for HY-∆TSI_max_. In the same sense, no differences were found in the other pairwise comparisons of the NIRS variables related to time-to-maximum or minimum values, hyperemia slopes, and half-time to recovery.

## 4. Discussion

The present study aimed to explore the microvascular adaptations of climbers compared to non-climbers by assessing the oxygen concentration and blood flow hemodynamics of the FDP muscle during a VOT. Our main findings revealed that climbers possessed specific microvascular adaptations that produced a greater and faster reactive hyperemia after the release of the occlusive pressure, and thus could be related to the ability to transport oxygen within the muscle tissue (see Figure 4 with a schematic summary of our findings and hypothesis).

Specifically, climbers showed a greater hyperemic slope of the blood flow compared to non-climbers. This is an important and novel finding since, to our knowledge, this is the first time that microvascular blood flow has been explored via photonics-based techniques during a VOT to assess potential exercise-training related adaptations, specifically in climbing. Other authors [32] have focused on functional and structural adaptations in climbers at the conduit arteries (i.e., brachial artery). These researchers have found that climbers did not possess a functional advantage compared to non-trained individuals based on their similar flow-mediated dilation (FMD) values [32]. However, when structural adaptations have been assessed, it has been reported that climbers possess enhanced brachial artery diameters at rest and maximal diameters during hyperemia [32]. Similar results of structural adaptations have also been observed in endurance trained individuals before [7]. These greater conduit artery diameters would facilitate greater blood inflow and oxygen transportation to muscles and could improve muscle endurance and function [4,32]. Yet, it is known that the magnitude of post-occlusion blood flow is not only regulated by conduit arteries but is also strongly governed by resistance vessels (i.e., small arteries and arterioles) [7]. In fact, blood flow dynamics between the conduit artery (explored via Doppler ultrasound) and microcirculation (explored via DCS) are markedly dissociated [33], which strengthens the need to also focus on the microcirculation blood flow kinetics. Therefore, the fact that the climbers in the present study experience faster blood flow increases during the reactive hyperemia could indicate better endothelial function (i.e., greater response to endothelial-dependent vasodilators) [9,22,34,35] or structural adaptations (i.e., vascular remodeling or angiogenesis) [7,36,37] at the microvascular level. In fact, other authors have observed that climbers possess greater capillary filtration than non-climbers [32], which could reinforce our hypothesis of peripheral structural and functional adaptations. Although both cases could be feasible (structural and/or functional adaptations), Bartlett et al. [33] hypothesized that greater vasodilatory responses could negatively impact the blood flow hyperemic slope, since greater dilatory responses during VOT-induced ischemia might create a volumetric void at the resistance vessels which has to be filled before pressure can be equalized across the arterial vascular tree. Therefore, although this should be further explored, it could be possible that the climbers’ greater hyperemic response was related to structural adaptations manifested as an increase in the number of resistance vessels (number of capillaries in parallel), which would allow a faster increase in the blood flow without the need for a large dilatory response. Indeed, previous research has observed improvements in capillary density in trained individuals and exercise training has been previously proposed as an important stimulus for improving muscular capillarity [7,36,37,38]. Given that climbing predominantly involves repeated isometric contractions with short resting periods in between, the ability to quickly re-establish blood flow (blood flow slope) during these rests could be crucial for performance. Therefore, blood flow slope during hyperemia should be a parameter considered by trainers and researchers. However, it is important to note that our interpretation may be limited by the lack of control over other potential confounding parameters, such as heart rate or arterial blood pressure during VOT.

Besides improved blood flow slope, climbers also presented a greater peak increase in O_2_Hb and a greater decrease in HHb concentrations during the reactive hyperemia. Increases in O_2_Hb concentration have previously been related to the restitution of blood flow after ischemia and have been used as a marker of vascular reactivity and endothelial function [4]. Thus, we suggest that the greater peak concentration of O_2_Hb in climbers could result from the faster blood flow slope, which would have caused greater O_2_Hb accumulation during the hyperemic response. Considering both results, the climbers’ increased blood flow slope and greater O_2_Hb concentration might result in a greater oxygen transportation, which could aid performance in active scenarios where periods of ischemia and reperfusion are alternated, which is highly common in climbing [2,8,39]. Similar results have been observed in other studies, where whole-body aerobic fitness was positively related to the magnitude of O_2_Hb increases during the VOT hyperemia in the vastus lateralis [40].

At the same time, the fact that climbers presented a greater decrease in the HHb concentration could indicate that they possess better venous return compared to non-climbers, represented by a greater HHb clearance. The improved venous return could result from an elevation of the arterial blood pressure due to the greater blood flow slope, along with a greater venous compliance. In fact, although most of the studies have focused on exercise-training related adaptations at the arterial level, there is evidence that exercise training improves venous compliance and thus the ability to accommodate rapid increases in blood volume in the venous circulation [41,42]. In fact, venous compliance adaptations have been more prominently observed in the exercised limb and after interval exercise interventions [43], which would match our findings with locally (i.e., forearm) trained individuals and with the naturally intervallic intensity of climbing. In conjunction, our results could add support to the hypothesis of structural adaptations (i.e., a greater number of vessels in parallel and vessel diameter), which would allow a faster increase in blood flow and thus greater HbO_2_ accumulation, accompanied by reduced venous congestion [4,9,41,42]. In this regard, a greater venous return could facilitate the washout of metabolic byproducts during the rest periods between contractions. This would help to maintain a stable metabolic state in the muscle environment and enhance the transport of O_2_.

Despite the novelty of our research, this is not the first time that microvascular exercise training-related and climbing-specific adaptations have been explored via NIRS during a VOT. Previous studies have reported improved hyperemic responses in whole-body aerobically trained individuals compared to non-trained ones [34,35,40]. When focusing in climbing research, similar results have been found, an improved hyperemia kinetics has been observed in climbers compared to non-climbers, and these hyperemia-related adaptations have been associated with climbing performance as well [20,21,22]. However, preceding publications mainly focused on the TSI signal, since it is the most common parameter in the clinical and research fields [44]. In this regard, the most common adaptations in aerobic fit individuals are seen in the slope of the TSI during the hyperemic response [34,35] and in the half-time to recovery (HTR) in climbers [20,21,22,30]. In our research, no differences between groups were found for any of the two variables (HY-∆TSI_HTR_ and HY-∆TSI_slope_). These contradictory results could be attributable to some of the limitations of our research.

The first of these limitations is that our study comprised climbers and non-climber healthy active participants. We did not evaluate whole-body aerobic fitness, but most of our non-climber participants reported regular practice of endurance and/or non-grip-related strength sports. Although some studies have reported that microvascular adaptations might be exclusive to the trained limb [34], this is a controversial topic. In fact, in Montero’s et al. review [9], it was argued that endothelium-dependent microvascular adaptations might not be confined only to active muscle beds, since most of the studies that found differences between trained and untrained individuals examined endothelial function in the non-trained limbs. It is assumed that vascular adaptations are primarily driven by repeated shear stress [11]. In this regard, cardiac demands generated by active muscles could result in (1) augmented blood flow and shear stress to non-working vessels and (2) sympathetically mediated arterial vasoconstriction in the non-working muscles, which should also increase shear stress [11]. Therefore, it could be possible that our non-climber individuals also presented vascular adaptations in the forearm due to exercise training, even though their regular exercise practice did not involve the forearm. Then, the inclusion of active, instead of sedentary, individuals could have affected our results in comparison to other studies. A second limitation to point out is that although climbers included in the present study were characterized as advanced climbers, most of them were on the lower end of the redpoint grades included in the advanced climbers category (13 climbers below an 8a redpoint grade out of 17) [29]. Including a larger proportion of higher-performance-level climbers could have elicited greater differences between climbers and non-climbers and provide greater insight in climbing-related microvascular adaptations. Lastly, a third limitation is that although differences between groups in the present study could potentially result from specific climbing training, the nature of our cross-sectional research makes it impossible to establish a direct link between training and the observed microvascular responses. In fact, this limitation is shared with most of the discussed studies, and it remains an open question that needs to be addressed by longitudinal interventional studies.

Regardless of the differences found during the reactive hyperemia, no between-group differences were observed during the occlusion or ischemia phase of the VOT. The fact that blood flow was reduced at a similar level between groups might indicate that the amount of pressure utilized in our study exposed the participants to an ischemic stimulus of similar intensity (i.e., full occlusion). In this regard, similar minimum values of O_2_Hb and TSI were also achieved for climbers and non-climbers, along with similar maximum values for the HHb concentration during the occlusion phase, which could support the rationale of similar ischemic stimulus among participants. In addition, we did not observe any differences in the slope of the NIRS signals during the occlusion phase of the VOT. It is thought that when supra-systolic tourniquet pressures are applied, the blood inflow and outflow become fully restricted, ensuring a closed circulatory compartment distally to the tourniquet. Under this circumstance, the oxygen consumed by the muscle determines the accumulation of HHb and the reduction in O_2_Hb concentrations. Consequently, the slopes of HHb, O_2_Hb, and TSI are usually interpreted as an indication of the ability of the muscle to extract or utilize oxygen (oxidative metabolism) [4,19,45]. Therefore, the absence of differences in the slopes of the NIRS signals during the OC phase in our study indicates that climbers exhibited similar oxygen consumption during the VOT, and thus no distinctive potential training-induced adaptations were observed. Previous research has reported greater deoxygenation slopes during ischemia in trained individuals when compared to non-trained ones [40,46]. It could be feasible that these greater deoxygenation slopes denote an increase in mitochondrial content within the muscle, since it is well known that mitochondrial biogenesis is triggered by exercise training [10]. In climbing research, it has also been reported that climbers possess a greater oxygen extraction capacity during active conditions [8,39,47], but to our knowledge, no previous reports exist about results during rest conditions. In addition, although previous climbing studies have reported adaptations in the oxidative metabolism of climbers, we could not observe such adaptations in our data. It could be possible that our measures of deoxygenation during the VOT were not sensitive enough to differentiate between the groups in the present study. Including a greater proportion of high-level climbers might have helped observing potential differences during the OC phase. However, based on our results, we believe that future climbing research would benefit from using other methods to assess oxidative metabolism, like those proposed by Ryan et al. [48,49]. In these studies, a series of short occlusions were used to assess oxygen consumption slopes after exercise. This technique has been cross-validated against the phosphocreatine recovery curve after exercise and is considered a great in vivo, cost-effective, and non-invasive means to assess oxidative metabolism [48,49]. Despite the acceptance of this methodology, it has not been used in climbing research before. In fact, most of the previous climbing studies utilizing NIRS during VOT have focused in the HTR parameter, which has been used as an indicator of the oxidative capacity of the muscle [20,21,22,30]. However, we believe that during hyperemia, the rapid and substantial increase in blood flow (which in our data averages 7–8 times fold the baseline value) and consequent increase in blood volume make it impossible to discriminate between oxygen consumption and transportation in this phase of the VOT. Therefore, in light of our results, where it seems that oxygenation and blood flow follow similar hyperemic dynamics, and in agreement with the most common interpretation of hyperemia in the literature, we think that HTR during the VOT at rest might better serve as an indicator of oxygen transportation capacity and endothelial function rather than a marker of oxidative metabolism. Nevertheless, simultaneous interpretation of blood flow and O_2_Hb dynamics in our study should be used carefully, since another of the limitations of our study is that NIRS and DCS probes were placed over slightly different portions of the muscle.

In conclusion, climbers seem to possess distinctive microvascular adaptations compared to active non-climber individuals, and thereby could grant better oxygen transportation within the muscle and could aid performance during sport practice. These distinctive microvascular responses manifested as faster increases in blood flow, greater concentrations of O_2_Hb and greater reductions in the HHb concentration during the reactive hyperemia of a VOT. Our results demonstrate the importance of exploring the microvascular blood flow along with the oxygen concentration in future studies that use a longitudinal approach to explore exercise-training-induced changes in microcirculation. Furthermore, trainers should consider that the rate and magnitude of reactive hyperemia during a VOT could potentially offer key insights into athletes’ ability to restore oxygen supply during short rests while climbing and, consequently, their potential performance.

## Figures and Tables

**Figure 1 bioengineering-11-00401-f001:**
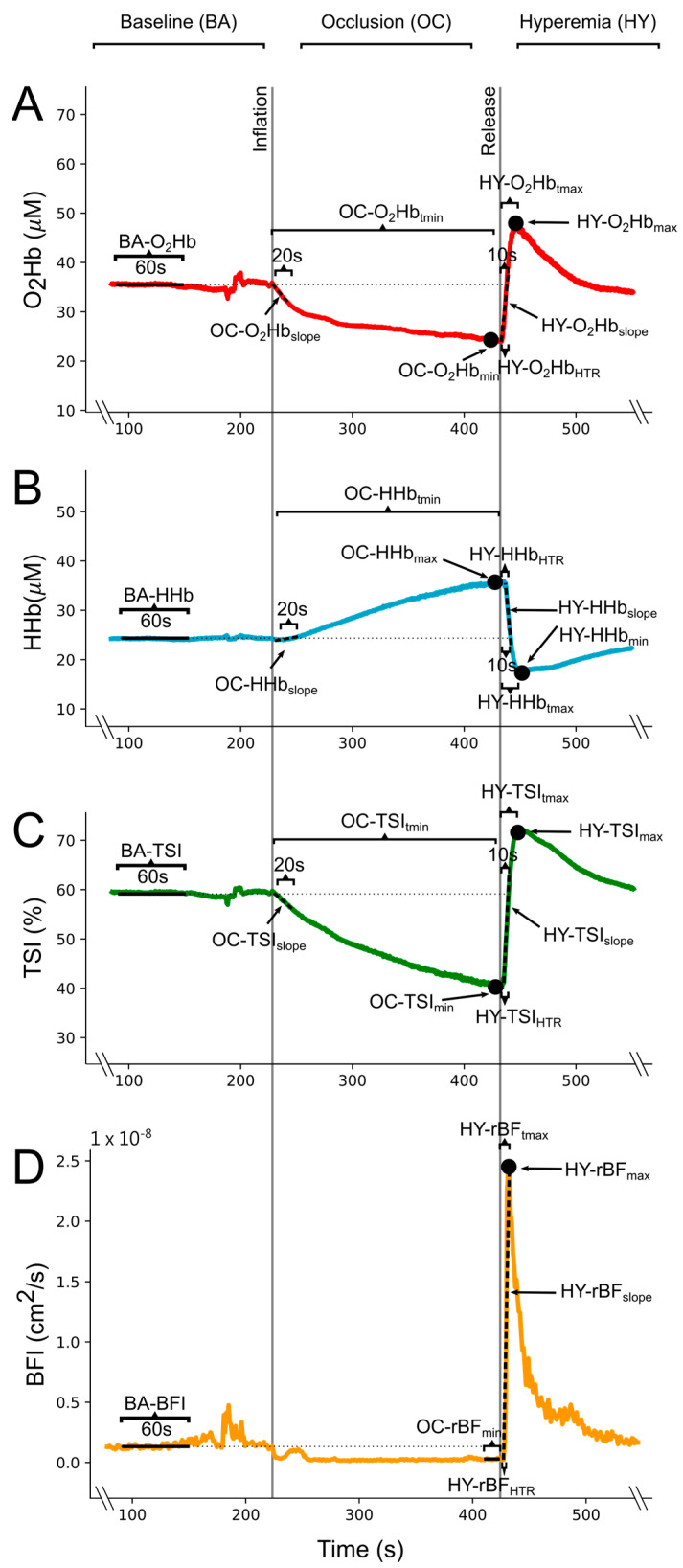
Example from a single participant of the (**A**) oxyhemoglobin concentration (O_2_Hb), (**B**) deoxyhemoglobin concentration (HHb), (**C**) tissue saturation index (TSI), and (**D**) blood flow index (BFI) changes during the vascular occlusion test (VOT). Vertical lines indicate the inflation and release of the pressure cuff, which were used to divide the test into the baseline phase (BA), occlusion phase (OC), and reactive hyperemia phase (HY). Calculated variables for each signal are schematically represented. Abbreviations: rBF: relative blood flow, tmin: time to minimum value, tmax: time to maximum value, HTR: half-time to recovery.

**Figure 2 bioengineering-11-00401-f002:**
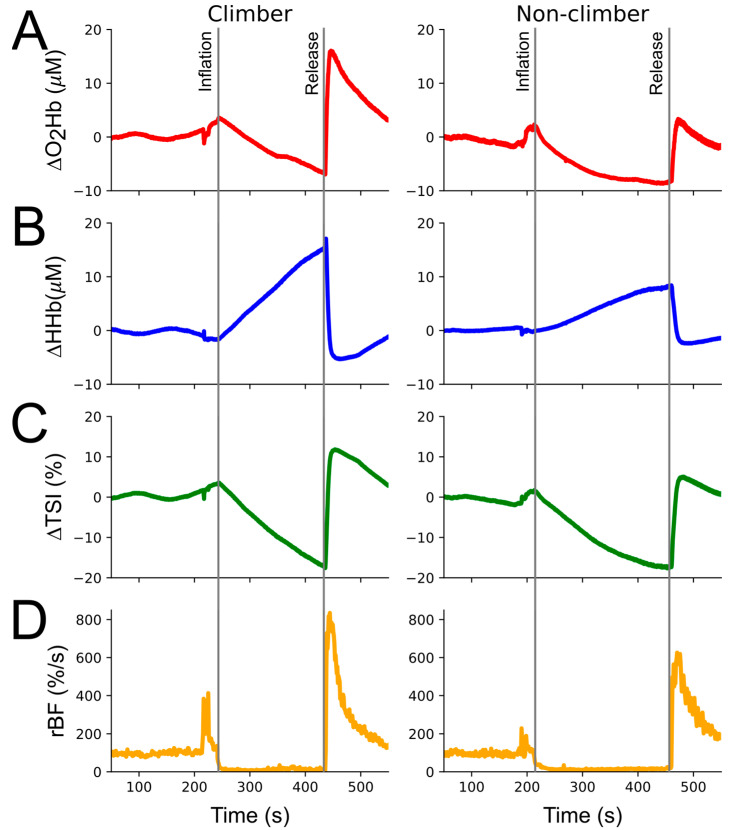
Normalized oxyhemoglobin (∆O_2_Hb, (**A**)), deoxyhemoglobin (∆HHb, (**B**)), tissue saturation index (∆TSI, (**C**)), and relative blood flow (rBF, (**D**)) example traces from a climber (**left side**) and a non-climber (**right side**) participant.

**Figure 3 bioengineering-11-00401-f003:**
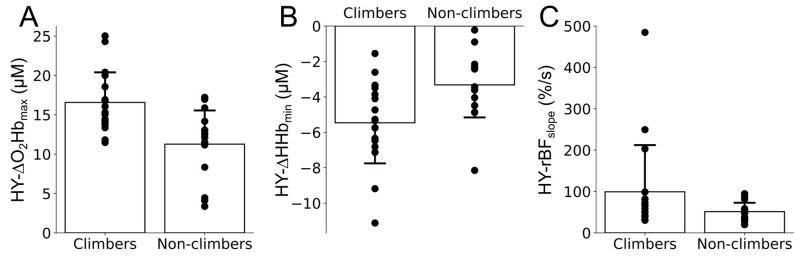
Group means and standard deviations, and individual subject’s values from the maximum delta hemoglobin concentration (HY-∆O_2_Hb_max_, (**A**)), the minimum delta deoxyhemoglobin concentration (HY-∆HHb_min_, (**B**)), and the relative blood flow slope (HY-rBF_slope_, (**C**)) during the hyperemia phase (HY) of the vascular occlusion test. Significant group differences were found for the three parameters (*p* < 0.05). When DCS-related variables during the HY phase were explored, no main group effect was found (*F* (1, 30) = 2.046, *p* = 0.130, *η*^2^*p* = 0.180, power = 0.468), indicating that groups presented similar blood flow dynamics when compared using the compound of DCS variables. Nevertheless, when pairwise comparisons were explored, climbers showed a greater HY-rBF_slope_ compared to non-climbers (*p* = 0.043, *d* = 0.573), indicating a faster increase in blood flow after the occlusive pressure release (Figure 3 and as can been seen in Figure 2 with sample data from each group). Despite this difference, similar HY-rBF_max_, HY-rBF_tmax_, and HY-rBF_HTR_ were found between groups.

**Figure 4 bioengineering-11-00401-f004:**
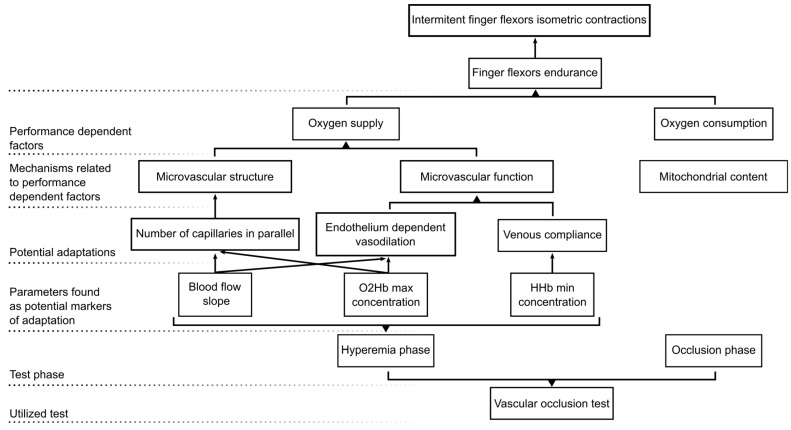
Schematic representation summarizing our study methodological approach, main findings, and hypotheses.

**Table 1 bioengineering-11-00401-t001:** Participant’s characteristics.

	Climbers	Non-Climbers
N	17	15
Age (years)	28.28 ± 6.0	25.83 ± 3.8
Biological sex	8 male/7 female	8 male/7 female
Height (cm)	170.5 ± 9.3	168.9 ± 8.4
Weight (Kg)	64.1 ± 9.6	65.8 ± 7.4
Arm span (cm)	174.8 ± 10.3	170.2 ± 10.3
Forearm perimeter	27.2 ± 2.4	25.7 ± 1.8
Forearm skinfold (mm)	1.7 ± 0.5	2.7 ± 1.3
Climbing experience (years)	8.6 ± 7.7	-
Climbing/training session per week	8.0 ± 0.9	
IRCRA grading scale	20.4 ± 3.2	-

**Table 2 bioengineering-11-00401-t002:** Group means and standard deviations and statistical results.

	Climbers	Non-Climbers
BA-O_2_Hb (µM)	43.6 ± 10.7	39.1 ± 11.7
BA-HHb (µM)	24.9 ± 4	19.1 ± 4.1
BA-TSI (%)	63.3 ± 3.9	66.4 ± 4.6
BA-BFI (cm^2^/s)	1.70 × 10^−9^ ± 9.12 × 10^−10^	1.40 × 10^−9^ ± 4.25 × 10^−10^
OC-∆O_2_Hb_slope_ (µM/s)	−0.1 ± 0.1	−0.1 ± 0.1
OC-∆HHb_slope_ (µM/s)	0.1 ± 0	0.1 ± 0
OC-∆TSI_slope_ (%/s)	−0.1 ± 0	−0.1 ± 0.1
OC-∆O_2_Hb_min_ (µM)	−6.5 ± 6.4	−8.6 ± 5.4
OC-∆HHb_max_ (µM)	14.9 ± 4.1	15 ± 6
OC-∆TSI_min_ (%)	−15.4 ± 8.5	−20.3 ± 7.8
OC-rBF_min_ (%)	12.3 ± 7.1	12 ± 4.1
OC-∆O_2_Hb_tmin_ (s)	−22.2 ± 43.6	−3.9 ± 7.8
OC-∆HHb_tmax_ (s)	−3.5 ± 12.8	−2.8 ± 8.8
OC-∆TSI_tmin_ (s)	−5.7 ± 13.9	−2.3 ± 4.5
**HY-∆O_2_Hb_max_ (µM)**	**16.6 ± 3.9**	**11.3 ± 4.4**
**HY-∆Hhb_min_ (µM)**	**−5.5 ± 2.4**	**−3.3 ± 1.9**
HY-∆TSI_max_ (%)	11.3 ± 4.2	9 ± 5.4
HY-rBF_max_ (%)	873.7 ± 440.3	712.4 ± 151
HY-∆O_2_Hb_tmax_ (s)	14.4 ± 4.2	16.3 ± 4.6
HY-∆HHb_tmin_ (s)	32 ± 7.8	33.9 ± 9.1
HY-∆TSI_tmax_ (s)	22.1 ± 6.1	24.5 ± 5.5
HY-rBF_tmax_ (s)	13 ± 6.5	15.6 ± 5.9
HY-∆O_2_Hb_slope_ (µM/s)	1.7 ± 0.6	1.6 ± 0.7
HY-∆HHb_slope_ (µM/s)	−1.4 ± 0.5	−1.3 ± 0.6
HY-∆TSI _slope_ (%/s)	2 ± 0.6	2.2 ± 1
**HY-rBF_slope_ (%/s)**	**100.8 ± 117.9**	**52.2 ± 22.7**
HY-∆O_2_Hb_HTR_ (s)	4.8 ± 1.8	5.3 ± 1.9
HY-∆HHb_HTR_ (s)	7.1 ± 3.1	7.6 ± 2.7
HY-∆TSI_HTR_ (s)	5.9 ± 2.4	6.5 ± 2.1
HY-rBF_HTR_ (s)	5.4 ± 4.3	4 ± 2.1

Notes: Statistically significant results are highlighted in bold. Abbreviations: BA: baseline phase; OC: occlusion phase; HY: reactive hyperemia phase; O_2_Hb: oxyhemoglobin concentration; HHb: deoxyhemoglobin concentration; TSI: tissue saturation index; BFI: blood flow index; rBF: relative blood flow; tmin: time to minimum value; tmax: time to maximum value; HTR: half-time to recovery.

## Data Availability

The data are available at https://doi.org/10.5281/zenodo.10893717.

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
