# Peer review of "Assessment of Microvascular Hemodynamic Adaptations in Finger Flexors of Climbers"

_bioengineering, 2024, doi:10.3390/bioengineering11040401_

Round 1

Reviewer 1 Report

Comments and Suggestions for Authors

Overview

The authors conducted a cross-sectional study comparing blood flow dynamics in climbers vs non-climbers.  Subjects were matched for age, sex, stature and body nass.  To qualify as a climber, specific terms were required apriori.  A non-invasive occlusion-reperfusion test was conducted and examined hemoglobin and deoxyhemoglobin values using NIRS and relative muscle blood flow using diffuse correlation spectroscopy.  The authors reported that during the baseline and occlusion phases, the hemodynamics were similar for climbers and non-climbers.  During the hyperemic phase post occlusion, climbers had a greater change in oxyhemoglobin (increase), deoxyhemoglobin (decrease), relative blood flow (nearly double) compared to the responses in the non-climbers.  The authors concluded that the observed differences were training-induced changes and would support the musculature and endurance needed during climbing.

Main Concerns

The paper appears to provide novel findings and is well written, for the most part.

The authors emphasize training adaptations as the cause of the group differences.  However, with all sports activities, there is a degree of natural selection for those who participate and continue to participate.  Some of that is due to natural or genetic differences.  This should be mentioned as a possibility, not just training adaptations, which were not tested in the study design.  Some specifics are provided below.

Is it possible that some of the change post occlusion is due to greater extraction of O2 by the forearm muscle?  Based on data in Table 1, climbers appear to have more forearm muscle mass.  With their training, I’d anticipate the mitochondrial density of the muscle is also greater.  I see this mentioned in lines 427-435.  Is this also complicated by potential differences in forearm muscle mass, regardless of muscle mitochondrial density? A specific comment is included below re muscle mass.  

While generally well written, proofreading is needed for several grammar issues.  Examples include  Lines 95-96 “…a rapid an (sp) great increase…,” Lines 102-103 “…this kind of measures (sp)…”

Were the mean heart rate and blood pressures measured during the three phases?  Presumably, HR and BP would be the same between the cohorts at baseline and during occlusion.  However, could a difference in the central response during the hyperemic phase have an influence on localized blood flow changes and group differences?  If HR and BP were not measured, this should be mentioned as a limitation; however, I would expect the authors could find a reference or two to refute the central response as being much of a factor.

Specific Issues

Line 23 “could indicate training-induced structural and functional adaptations…:” I thought this was a cross-sectional study using a comparison between groups.  How can the authors conclude “training-induced” without longitudinal data for a treatment effect?  Inserting “potential” before “training-induced” is recommended.

Table 1: Were forearm circumference and/or forearm skinfold different between the two groups?  If so, did the authors consider standardizing the main outcome variables for a proxy for forearm muscle mass?  If this is a confounding variable, see DeFreitas JM, et al. A comparison of techniques for estimating training-induced changes in muscle cross-sectional area. J Strength Cond Res. 2010 Sep;24(9):2383-9. doi: 10.1519/JSC.0b013e3181ec86f3.  The authors likely have the data to make this estimation.

Line 385 “However, the focus on these preceding publications mainly fell in the TSI signal,…:”  Suggest changing “fell” to “addressed.” As written, “fell” may confuse the reader with decreasing TSI signal.

Line 392 “In first place, our study comprised climbers…” In first place with respect to what?  This seems out of context.

Line 444 “…non-climber individuals, which could grant better oxygen transportation…:” Because the phrase following “which describes the climbers, this would read accurately as “...individuals, and thereby have better…”

Line 444-445 “…oxygen transportation to the muscle…:” Do the authors mean “within the muscle?”

Lines 445-446 “These microvascular adaptations…:” If I understand this, adaptations should be “acute responses.”  Adaptations typically are a result of conditioning or training, which was not tested in this study.  The authors measured acute responses for a group comparison, not longitudinal changes with a treatment.

Line 448 “Our results claim the importance…:”  How about “Our results demonstrate…?”  The claim made by an inanimate entity (results) doesn’t make sense.

Lines 449-450 “..when exploring the microcirculation function to assess exercise-training adaptations:” To help move forward from descriptive to experimental research, how about concluding with something like, “…in future studies that use a longitudinal approach to explore exercise-training induced changes in microcirculation.”

Comments on the Quality of English Language

Minor edits mentioned in review to authors.

Author Response

Dear Reviewer,

Thank you very much for dedicating your time to reviewing our manuscript. We have carefully considered all of your comments and have provided responses to each of them. You can find our detailed responses in the attached file.

We sincerely appreciate your valuable feedback and look forward to hearing your thoughts on our revisions.

Best regards,

Authors

Reviewer 2 Report

Comments and Suggestions for Authors

Dear Editor,

Thank you for the opportunity to review the manuscript "Assessment of microvascular hemodynamic adaptations in finger flexors of climbers" submitted to the Bioengineering journal. The manuscript examines the differences in microvascular function between climbers and non-climbers through the assessment of oxygen concentration and blood flow hemodynamics in the forearm during a vascular occlusion test (VOT). The main findings indicate that climbers show superior microvascular adaptations, as evidenced by a faster increase in blood flow and greater changes in oxyhemoglobin and deoxyhemoglobin concentrations during the reactive hyperemia phase of the VOT, compared to non-climber active individuals.

Overall, the manuscript is well-written, the methodology is sound, and the results are presented clearly and comprehensively. The findings contribute to the understanding of the physiological mechanisms underlying the enhanced performance of climbers and could have important implications for training and performance monitoring in the sport.

General comments:

  1. The authors should provide more context and background information in the introduction to establish the importance of microvascular function in the context of climbing performance. This would help the reader understand the rationale and significance of the study.
  2. The discussion could be expanded to provide a more in-depth interpretation of the results, particularly in relation to the existing literature on microvascular adaptations in trained individuals and climbers. This would help the reader better understand the implications of the findings.
  3. The authors should address the potential limitations of the study, such as the inclusion of active, rather than sedentary, non-climbers, and the relatively low performance level of some of the climbers. This would help the reader assess the generalizability and interpretation of the results.
  4. The authors could consider incorporating a schematic or visual representation of the proposed mechanisms underlying the observed microvascular adaptations in climbers. This would help the reader better understand the physiological processes involved.
  5. The authors should discuss the potential practical applications of the findings, such as implications for training, performance monitoring, and injury prevention in the sport of climbing.

Specific comments:

  1. Introduction:
    • The authors could provide more details on the potential mechanisms underlying the microvascular adaptations in climbers, such as the role of repetitive hemodynamic stimuli and shear stress during climbing. This would help the reader understand the physiological basis for the observed adaptations.
  2. Materials and Methods:
    • The authors could provide more information on the inclusion criteria for the climbers, such as the minimum required climbing experience and performance level. This would help the reader assess the representativeness of the climber sample.
    • The authors could also consider including more details on the physical characteristics of the participants, such as their training history and body composition. This would provide additional context for interpreting the results.
  3. Results:
    • The authors should discuss the potential reasons for the lack of differences in the oxygen consumption parameters (OC-∆O2Hbslope, OC-∆HHbslope, OC-∆TSIslope) during the occlusion phase of the VOT. This would help the reader understand the specific adaptations observed in the climbers.
  4. Discussion:
    • The authors could elaborate on the potential structural and functional adaptations at the microvascular level that could explain the superior hemodynamic responses observed in climbers, and how these adaptations might contribute to climbing performance. This would provide a more comprehensive interpretation of the findings.
    • The authors could discuss the implications of their findings for the interpretation of the half-time to recovery (HTR) parameter during the VOT and its use as an indicator of oxidative capacity in climbers. This would help the reader understand the relevance of the findings in the context of previous research on climbers.

Overall, the manuscript presents an interesting and well-designed study that contributes to the understanding of the physiological adaptations underlying climbing performance. With the suggested revisions, the authors can further strengthen the impact and clarity of their findings.

Sincerely,

Author Response

(The authors gave the same response as above.)
